# Music Therapy for Gait and Speech Deficits in Parkinson’s Disease: A Mini-Review

**DOI:** 10.3390/brainsci13070993

**Published:** 2023-06-25

**Authors:** Leon Fan, Ellen Y. Hu, Grace E. Hey, Wei Hu

**Affiliations:** 1Computer Science and Molecular Biology, Massachusetts Institute of Technology, Cambridge, MA 02139, USA; 2The International Baccalaureate Program, Tampa, FL 33610, USA; 3Department of Neurology, The University of Florida, Gainesville, FL 32607, USA

**Keywords:** music therapy, gait disorder, speech disorder, Parkinson’s disease

## Abstract

Parkinson’s disease (PD) is a progressive central nervous system disease with a common motor symptom of gait disturbance in PD, which is more pronounced in the later stages. Although FDA-approved treatments, including dopaminergic pharmacotherapy, deep brain stimulation, and rehabilitation, have some benefits in improving gait dysfunction, a fair amount of advanced PD patients can develop a disability, social isolation, and high mortality and morbidity. Recently, clinicians and scientists have applied music to clinical therapy, namely music therapy. It has been used as a unique rehabilitation tool to improve PD-induced gait and speech disorders. Based on relevant studies in recent years, this paper reviews the published literature about music in treating gait disorders and speech problems in PD patients. Additionally, we discuss current studies’ limitations and emphasize the future potential research fields.

## 1. Introduction

Parkinson’s disease (PD) is a progressive central nervous system disease characterized by motor symptoms, including tremors, rigidity, bradykinesia, and gait disturbances. Significant non-motor symptoms, including depression and anxiety, are associated with Quality of Life (QoL) reduction [1,2,3]. It has low mortality but a high disability rate, with a prevalence rate of about 0.3% [3,4]. Older adults are at increased risk for PD, with a prevalence rate of 1~2% among individuals aged 65 or older and 3~5% in people over 85 [3,4]. The number of PD patients was approximately 8 million in 2021, and as influenced by global aging trends, the number of PD patients worldwide is expected to approach 10 million by 2030 [3,4].

It should be noted that gait disturbance is a common motor symptom in PD and is more pronounced in the later stages of the disease [4]. PD patients with gait disturbance always have slower gait velocity, decreased stride length with increased cadence, shuffling steps, and particularly freezing of gait, which is an episodic motor disturbance in PD that causes patients to be unable to maintain and/or initiate their locomotion despite their intention to walk. This is one of the main reasons for falls and mortality in PD patients [5]. On the other hand, PD patients may suffer from speech disorders. It has been reported that as many as 89% of PD patients have speech disorders with various motor activity problems regarding respiration, phonation, articulation, resonance, and prosody. FDA-approved treatments, including dopaminergic pharmacotherapy, deep brain stimulation, and rehabilitation, have some benefits in improving gait dysfunction but not for speech disorders [2,3,4,5,6,7,8]. Subsequently, these disturbances could decrease mobility with a high risk of falling and reduce the quality of life for people with PD. Further, it can lead to disability, social isolation, and high mortality and morbidity [4,5,6,7,8,9]. Recently, it has been reported that rehabilitation approaches, including physical therapy, occupational therapy, and speech therapy, can also be somewhat beneficial [3,4,5,6,7]. On the other hand, PD medications have side effects of dyskinesia and are sometimes ineffective in PD symptom control. Therefore, clinicians and scientists are working together to find an alternative intervention for gait disorders in PD [7].

Music is the art of sound in time and can express emotions and ideas through the elements of rhythm, melody, harmony, and color [10,11]. It has been reported that music can activate motor neurons through its rhythms and then make muscles contract, and body movements synchronize with the beat [12]. Specifically, when music stimulates the auditory processing area, it interacts with its adjacent motor area. Subsequently, the human body will show corresponding motor performance accordingly based on the music stimulation. As such, clinicians and scientists have applied music to clinical therapy, namely music therapy. More recently, music has been used as a unique rehabilitation tool to improve PD-induced gait disorders [11,12]. On the other hand, it has been reported that music therapy can improve speech in individuals with speech disorders such as stuttering, apraxia, and dysarthria. music therapy techniques such as rhythmic speech cueing, melodic intonation therapy, and vocal improvisation can help improve the timing, rhythm, and intonation of speech. music can also increase motivation and engagement in speech therapy, making it a more enjoyable and effective experience for individuals with speech disorders. Additionally, music has been found to positively affect mood and stress reduction, which can also contribute to improved speech production. Based on relevant studies in recent years, this paper reviews the published papers about music in treating gait disorders and speech problems in PD patients. We hereby discuss current studies’ limitations and emphasize the future potential research fields.

## 2. Materials and Methods

We performed a systematic review of articles discussing the management of music therapy in PD between January 2010 and December 2022. We examined data from 78 studies detailing the effects of music therapy on gait and speech deficits in PD. Clinical outcomes were measured by the suitable scale. Most studies addressing treatment response were randomized control studies. Specifically, the inclusion criteria were: scientific papers published in peer-reviewed scientific articles in English and research on music therapy for PD in PubMed between 2010–2022. Exclusion criteria: publications that were not accessible or not in English. In total, we included 46 relevant published papers in the analysis.

## 3. Music Therapy in PD Gait Disorder

Although music therapy (MT) has been widely used in various settings. It is still poorly evaluated in the medical care of neurological disorders. MT has two branches: active and passive. More and more clinicians and scientists conducted numerous prospective studies to assess the effects of MT in the neurorehabilitation of PD patients. An initial study reported that sixteen PD patients attended 13 weekly sessions of MT, and each session lasted 2 h. The Parkinson’s Disease Quality of Life Questionnaire (PDQL) was evaluated by a neurologist at the beginning and the end of the session. They identified a significant improvement in PDQL in these patients after every session. Recently, a Romanian pilot study investigated if music could potentially be an add-on therapy for PD patients who have been in a rehabilitation program [13]. Specifically, the goal of this study is to examine the effects of a rehabilitation program with physical therapy and music listening in people with PD. Sixteen PD patients attended a specific rehabilitation program with background music for two and a half hours daily for two weeks. They found that patients in the study group had greater improvements in mobility, the activity of daily living (ADL), emotional well-being, stigma, and bodily discomfort in the eight areas of life assessed by the Parkinson’s Disease Questionnaire (PDQ-39). Their study suggested that music therapy could improve QoL in PD patients. This approach combining rehabilitation therapy and music listening may be more feasible and widely acceptable for retired patients in urban areas. The local society could organize such programs for PD patients.

A Japanese group used a portable gait rhythm gram to examine the effect of music therapy on PD-related gait disturbance [14]. Specifically, 19 PD subjects with gait disturbance participated in this study. The scientists evaluated their gait speed and step length acceleration, cadence, and trajectory during walking tasks. They found an immediate improvement in gait speed, cadence, acceleration, and step length in tasks with music by a 10 m straight walking task test. Based on this study, the authors suggested that rehabilitation with music could be simple, cheap, noninvasive, nonpharmacological, and more suitable for rehabilitating PD-related gait disturbance every day.

A group of German movement disorder specialists recently used closed-loop musical feedback (musification) to assess its effect on gait parameters and arm swing amplitude in patients with PD by Gait kinematic [15]. They found that musification could induce a large and bilateral increase in arm swing. This benefit is more prominent on the more affected side of the patient. Moreover, this musification could increase the stride length and symmetry of the arm swing. Their results indicated that musification immediately affects gait kinematics and arm swing in PD. This closed-loop musical feedback could be an effective technique to help gait issues in patients with PD.

A European neurologist team studied personalized-music-based gait rehabilitation in PD [16]. Specifically, 45 PD patients had an outdoor rehabilitation program with the BeatWalk application for one month (30 min/day, five days/week). After the program, the gait parameters of the studied subjects were improved in the walk test. This study suggested that BeatWalk is an easy-to-use and safe musical application for PD patients with gait rehabilitation.

In 2023, a Chinese research group studied the effect of music-based movement therapy on the GOP in PD patients by prospective, evaluator-blinded, randomized controlled trial [17]. They randomly divided 81 participants into a music-based movement therapy group (MMT), exercise therapy group, and control group. MMT was performed five times (1 h at a time) every week for four weeks. They found that MMT improved the max flexion of the knee in stance, the max hip extension, the flexion moment of the knee in stance, the double support time, the cadence, the comprehensive motor function, and the FOG-Q gait disorders. Their study indicated that MMT could improve comprehensive motor function in PD patients with FOG.

Although numerous clinical studies about music therapy in PD have been completed, few studies explored its potential mechanism [18,19,20,21,22,23,24]. The effect may be related to music cueing and its benefit on the activity of basal ganglia [25,26,27,28,29,30,31,32]. Recently, an Italian group examined the role of the cerebellum in gait performance improvement in PD people receiving neurologic music therapy. A standard EEG during gait was also recorded. However, the results have not been published yet [30,31,32,33,34].

Previously, most research studies focused on the application of RAS in the treatment of PD gait symptoms. A recent pilot study investigated the role of the combination of three neurologic music therapy (NMT) sensorimotor techniques in PD gait [35]. Specifically, the 55 PD patients were divided into the experimental group with 30 subjects and the control group with 25 subjects. They included PD patients with Hoehn and Yahr, stages 2 or 3, who can walk independently without aid. Those patients had stable medical therapy. Additionally, the authors measured the gait, stability, and balance parameters, including Optoelectrical 3D Movement Analysis, System BTS Smart, and Computerized Dynamic Posturography CQ Stab. In the experimental group, the subjects had music therapy four times a week for 4 weeks. This combined music therapy includes therapeutic instrumental music performance (TIMP), pattern sensory enhancement (PSE), and RAS. In the control group, the subjects stayed active and performed daily life activities. They found that combining the three NMT sensorimotor techniques can improve the spatiotemporal gait parameters and gait dysfunction in PD. In addition, the stability tests with eyes closed identified the improvement of proprioception (the sense of body position and movement).

Pacchetti C. et al. performed a randomized, controlled, blinded study to explore the benefit of active MT on motor and emotional functions in PD subjects [36]. The patients received weekly sessions of MT and physical therapy (PT). In total, thirty-two PD patients were randomly assigned to two groups. The authors evaluated the Unified Parkinson’s Disease Rating Scale, the Happiness Measure, and the Parkinson’s Disease Quality of Life Questionnaire. MT sessions included voice exercises, choral singing, and rhythmic body movements. PT sessions consisted of specific motor tasks, passive stretching exercises, and balance and gait improvement strategies. By the Unified Parkinson’s Disease Rating Scale, MT had a significant benefit on motor improvement, especially in bradykinesia items. Moreover, these patients had improvements in emotional functions, activities of daily living, and quality of life [37,38]. The authors concluded that MT is effective and can be used as a new method for PD rehabilitation programs. Moreover, Calabro et al. [39] enrolled PD patients with treadmill gait training and RAS for 8 weeks. Then, they measured clinical, kinematic, and electrophysiological effects and found that comparable to treadmill training only, the combination therapy could have a greater improvement in Functional Gait Assessment, Tinetti Falls Efficacy Scale, Unified Parkinson’s Disease Rating Scale, and overall gait quality index. In addition, they found that the RAS and gait training induced a stronger EEG power increase within the sensorimotor rhythms. A European cooperation group [40,41,42,43,44,45] noticed that the beneficial effect of music therapy on PD gait is highly variable among patients, which might be related to patients’ ability to synchronize their movements to a beat. Dalla Bella et al. tested this possibility by studying 14 PD patients and found some patients showed a positive response to RAS, and other subjects had no response. They further identified that the synchronization performance in hand tapping and gait tasks could predict a positive response to RAS. This result suggested that sensorimotor timing skills may play a role in the success of RAS in PD. In addition, it has been established that PD patients have cognitive, psychiatric, and mood disorders. Music has been successfully applied to these non-motor symptoms. Morris administered a survey study on 19 PD patients with music therapy and found that these subjects may receive more pleasure and value from music therapy. They were more adherent to therapy and had a better quality of life for people. In addition, it has been reported that music therapy has beneficial effects on cognitive function in PD patients. Therefore, this therapy can improve the mood and cognitive sphere in PD patients and subsequently benefit the gait dysfunction and quality of life in these patients [46].

The articles that address effects of music therapy on gait dysfunction in Parkinson’s disease are listed in Table 1.

## 4. Music Therapy in Speech Disorder

It should be noted that PD patients can exhibit speech production impairments. However, there are very few studies that have investigated whether or not music intervention has any benefit on vocal abilities in PD. Recently, Haneish E. et al. evaluate the effect of a group voice and singing on speech, singing, and depressive symptoms in PD patients [38]. Specifically, ten patients received a one-hour intervention once a week for 20 weeks. The authors applied the measurements of the KayPentax Multi-Dimensional Voice Program, the Voice Handicap Index (VHI), and the Montgomery and Asberg Depression Rating Scale (MADRS). They tested speech and singing quality at baseline and after 20 weekly sessions. They found that singing quality outcomes, voice range, and the VHI physical subscale significantly improved. Additionally, speaking quality outcomes remained the same during the study. On the other hand, Pacchetti C. et al. [36] examined the effects of a music Therapy Voice Protocol (MTVP) on speech intelligibility, the maximum duration of sustained vowel phonation, vocal fundamental frequency, vocal intensity, maximum vocal range, vocal fundamental frequency variability, and mood in 4 female PD patients. However, the authors found no significant improvement in acoustic variables in these subjects. Additionally, Butala A et al. [48] assessed the effects of weekly group singing on PD patients’ objective vocal and motoric function, cognition, mood, self-efficacy, and quality of life. They found a significant improvement in the Cookie Theft picture description, minimal reading volumes, minimal loudness on Rainbow passage reading, Emotional Well-Being, and Body Discomfort domains of the PDQ-39. Their study indicated that weekly group singing could improve some aspects of conversational voice volume and quality of life in PD.

## 5. The Potential Mechanism of Music Therapy

It has been established that neurologic music therapy has a tool for auditory entrainment of motor function and can improve clinical outcomes in PD patients. However, the exact mechanisms are not largely known yet. Tomaino C. reviewed the related paper to bridge science and music-based interventions and suggested the cerebellum’s role, along with other subcortical systems, in pre-learned motor schema modulation during neurologic music therapy [40]. Brancatisano O. et al. [41] proposed the Therapeutic Music Capacities Model that links individual properties of music to therapeutic mechanisms and its benefit on cognitive, psychosocial, behavioral, and motor function. Furthermore, Buard I. et al. [42] utilize neuroimaging and neurophysiological tools to explore the potential mechanisms in music therapy for motor impairments. They found that 5 weeks of NMT had beneficial effects on fine-motor function in PD. Importantly, during a cued-finger-tapping test, MATLAB spectral analysis showed an increase in evoked power in the beta range in the primary motor and auditory cortices task. This result suggests that an activity coupling in those two areas was related to the simultaneous activation. The study results suggested that musical interventions may influence cortical activity. Specifically, after music therapy, the auditory and motor cortices would be more strongly functionally connected, which can be observed by neuroimaging and neurophysiological tools. Additionally, Braunlich K. et al. [44] investigated the neural systems underlying the music therapeutic effect by scanning PD patients and age-matched healthy controls using functional magnetic resonance imaging (fMRI). All subjects performed rhythmic motor behavior with and without simultaneous auditory rhythmic cues. Using spatial independent component analysis (ICA) and regression, the authors identified task-related functional connectivity networks. They found greater inter-network connectivity between the auditory and executive control networks in PD subjects (Figure 1). This finding may be responsible for cortico-cerebellar network activity modulation. In addition, a secondary EEG analysis of a randomized clinical trial in PD patients suggested that the cerebellum mediates the reshaping of sensorimotor rhythms and front-centroparietal connectivity concerning specific gait-cycle phases [49].

## 6. Discussion

Numerous studies have focused on the combination of RAS and neurologic music therapy (NMT) sensorimotor techniques in the improvement of spatiotemporal gait parameters and gait dysfunction in PD. On the other hand, music-based therapies can increase socialization and may have mood and cognitive benefits, and song-based therapies may improve voice quality and respiratory and swallow function. in PD. Additionally, it is critical for scientists to study the role of different musical elements, including frequency, tempo, melody, and playback volume, in the clinical outcome of PD patients. Recently, more and more PD patients began to receive telemedicine rehabilitation services. Music therapy could also be used in that field. Telemedicine can be a useful tool in managing PD, especially for individuals who live in remote or underserved areas, or who have difficulty traveling to in-person appointments. Through telemedicine, healthcare providers can conduct virtual consultations and assessments, monitor symptoms and medication use, adjust treatment plans, and provide education and support to patients and their families. Telemedicine technology can also be used to deliver speech and physical therapy interventions, such as LSVT (Lee Silverman Voice Treatment) and BIG therapy, which effectively improve speech and motor function in individuals with PD. Additionally, telemedicine can provide psychological support and counseling to individuals with PD and their caregivers. Thus, telemedicine can deliver music therapy for PD patients remotely.

This review focuses on the gait and speech dysfunction in PD as these symptoms cannot be well controlled by medications in many PD patients. We included important related literature published in PubMed but not in other databases. Additionally, since some studies might use different connotations of the term “music therapy”, we could not identify these publications to some extent. Thus, it is likely that not all clinical studies were analyzed in this review. Our group will continue to search more databases and provide another updated review for the readers in the near future.

## 7. Conclusions

Taken together, it has been established that music therapy can help PD patients. Music therapy has been shown to be effective in improving gait in individuals with neurological conditions such as PD. Research studies have found that using rhythmic auditory stimulation (RAS), which involves synchronizing movement with an external beat, can improve walking speed, stride length, and overall gait quality. Additionally, music has been found to positively affect mood and motivation, which can also contribute to improved gait. It should be noted that all these published studies are short-term research projects with a period of 1–3 months. Further studies are needed to evaluate this therapy’s long-term effects and compare it with other established exercise techniques.

## Figures and Tables

**Figure 1 brainsci-13-00993-f001:**
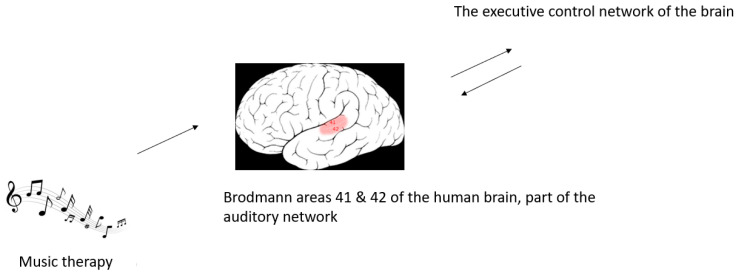
Music therapy can induce greater inter-network connectivity between the auditory network and the executive control network in Parkinson disease.

**Table 1 brainsci-13-00993-t001:** Important articles that address the effects of music therapy on Parkinson’s disease (2021–2023).

Author	Country	Year of Publication	Sample Size	Type of Study	Results
Gondo E. et al. [14]	Japan	2021	19	Intervention	Rhythmogram can significant improved acceleration, gait speed, cadence, and step length in PD patients
Fodor D. et al. [13]	Romania	2021	32	Pilot study	Listening to music combined with a multimodal rehabilitation program centered on physical therapy may be beneficial for the patient’s quality of life through improvements in PDQ-39.
Mainka S. et al. [15]	Germany	2021	30	Intervention	Closed-loop musical feedback has an immediate effect on arm swing and other gait kinematics in PD.
De Cock C. et al. [16]	France	2021	44	Intervention	After the program, patients improved their gait parameters in the six-minute walk test without musical stimulation
Machado Sotomayor M.J. et al. [12]	Spain	2021	N/A	Review	It was concluded that music therapy programs can achieve improvements in various areas of patients with PD.
Wu Z. et al. [11]	China	2022	N/A	Review	Music therapy is an effective way to treat the gait disorders caused by PD.
Li K. et al. [17]	China	2022	81	RCT	Music-based movement therapy (MMT) could relieve the freezing of gait and improve the quality of life for patients with PD.
Naro A. et al. [47]	Italy	2023	50	RCT	Rhythmic Auditory Stimulation (RAS) improved gait performance in PD.

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
