# Peer review of "Music Therapy for Gait and Speech Deficits in Parkinson’s Disease: A Mini-Review"

_brainsci, 2023, doi:10.3390/brainsci13070993_

Round 1

Reviewer 1 Report

Comments and Suggestions for Authors

The review highlights on the benefit of music in the treatment of gait and speech disorders in PD patients. Music engages so many neural systems that it would be important to pinpoint the brain correlates involved in improvement of PD. However, this review gives us the impression that not many neuromechanistic based studies have been conducted in this area.

The review though well written has still room for improvement. A revision of the manuscript may be considered.

1.       Coverage of literature has been restricted to recent years and the table is limited to studies conducted only between 2021-2023. Basically, this is an underrepresentation of the studies conducted in this field. For example, the study by Dalla Bella (Sci Rep) 2017 showed that gait improvement via rhythmic stimulation in PD to be linked to rhythmic skills. Literature available on oldest extant form of music and ancient texts on therapeutic potential of music in PD may also be cited..

2.       The authors may consider of including fMRI studies on task-related identification and assessment of functional network connectivity in addition to study on gait kinematics, arm swing and motor functions. Besides clinical measures, functional studies based on measuring brain activity by conventional EEG recording analysis and electrophysiology may be discussed as well. It would be also interesting to know if there have been any biomarker-related studies in this field.

3.       The authors may also look for studies that combined music with other non-music interventional components in PD. For example, are there studies that have used music/ dance or music/imagery in context of PD? There is a study by Calabro et al, 2019 where they have used treadmill along with music for PD patients. It would be interesting as well to know if there are reports on use of different musical elements, such as frequency, tempo, melody and playback volume in PD and if these musical elements have influence on the outcome measures in PD patients. The prospects of melodic intonation/melodic therapy/choir singing and various music-based intervention (MBI) strategies in PD may be discussed further.

4.       There has been a significant progress in the application of rhythmic auditory stimulation (RAS) and combination of neurologic music therapy (NMT) sensorimotor techniques in improvement of spatiotemporal gait parameters and gait dysfunction in PD. Song-based therapies for PD may improve voice quality, respiratory function and swallow function. Music-based complementary therapies can increase socialization and may have mood and cognitive-boosting benefits in PD. These areas were overlooked. Despite the number of reports available, there is a scarcity of well-powered data and appropriately designed clinical trials on music in PD. The authors are encouraged to discuss the pitfalls and strength of some studies from this point of view.

5.       The recent “Parkinsonics” study of ‘group singing’ as a therapeutic intervention for motor and nonmotor symptoms and improvement of speaking volume in PD from Alexander Pantlyat’s group (Center for Music and Medicine, Johns Hopkins) (Butala et al, 2022) may be included.

6.       A diagrammatic representation of brain circuitry underlying neural and physiologic responses to music in PD may be included to make the review more interesting.

7.       The bibliography should be updated and numbering to be formatted properly.

8.       The title is broad but if the music therapy is purposely restricted to gait and speech disorders in PD, it should be reflected in the title.

Author Response

Dear Editors,

Thank you for your decision letter and your advice on our manuscript entitled “Music therapy for gait and speech deficits in Parkinson dis-ease: A Mini-review”. We have revised the manuscript according to all feedback. All of the changes are highlighted in red within the revised manuscript. In addition, point-by-point responses to each of the comments are listed below.

We hope that the revision is acceptable for publication in your journal and we thank you and the reviewers for the thoughtful comments.

We look forward to hearing from you soon, 

Sincerely,

Wei Hu MD PhD

Reviewer 2 Report

Comments and Suggestions for Authors

Dear Authors,

Your paper has great potential, but a lot of work should be provided in order to have an outstanding study. Therefore, I advise:

- in the Abstract to write "Music" in lowercase and to add more information on the matter.

- the keywords are missing.

- the references throught the text must be in brackets, i.e., "[1]."

- support every statistic with references, i.e., "Older adults are at increased risk for PD, with a preva- lence rate of 1~2% among individuals aged 65 or older and 3~5% in people over 85. The number of PD patients was approximately 8 million in 2021" and "It has been reported that as many as 89% PD patients have speech disorders with various motor activity prob- lems in terms of respiration, phonation, articulation, resonance, and prosody" among other examples.

- the authors must state the material and method for their design, how the studies were selected, the medical platforms used (online libraries), and inclusion and exclusion criteria.

- chapter and subchapter numbers must be added.

- the article must be redesigned in order to have at least two figures and two tables.

- the conclusion is too long 

- more references are needed. 

- English need an extensive revision.

 Overall, it is a good text, but the authors need to add more value to it.

Comments on the Quality of English Language

Author Response

(The authors gave the same response as above.)

Reviewer 3 Report

Comments and Suggestions for Authors

The authors reported a review on Music Therapy in Parkinson’s Disease. I have some comments to the authors: 

-       The authors should better address why they focus just on gait and speech disorders.

-       The authors should specify how they perform the review, indeed the manuscript lacks of the methods section. Please add it with all the relevant info.

-       The paper needs to be more structured. Alongside with the methods section, the authors should better describe the results, without any comments. The comments should be limited in the discussion part (to be added). The discussion should also include limitations and strengths of the study.

-       “More and more clinician and scientists conducted numerous prospective studies to assess the effects of MT in the neurorehabilitation of PD patients.” That’s not an appropriate sentence for a scientific article, you should state how many papers have been published thanks to your review (this is one the goal of a review).

-       “We think this approach combining rehabilitation therapy and music listening may be more feasible and widely acceptable for retired patients in urban areas. The local society could organize such programs for PD patients”. This consideration and other personal views by the authors should be stated in the discussion.

-       “Recently, more and more PD patients began to receive telemedicine..” the authors should move this part to the new paragraph of the discussion, it is better to explore data in the discussion rather than at the end of the study.

-       “On the other hand, every week, the author volunteers at an assisted living …” I did not get the last paragraph of the conclusion and why the reader should be interested in this anecdote. The manuscript should be limited to the analysis of the data in the literature. 

Author Response

(The authors gave the same response as above.)

Round 2

Reviewer 1 Report

Comments and Suggestions for Authors

Thank you for incorporating the suggestions.

Author Response

Dear Editors,

We would like to express our sincere gratitude to the reviewers for their constructive and insightful comments.

Point-by-point responses to each of the comments are listed below.

Reviewer 1: Thank you for incorporating the suggestions.

Thank you very much for your comments and help!

Reviewer 2: Congratulations for your article!

The article is relevant for the literature and also is significant for the field and for the readers.

We appreciate your help and support.

Reviewer 3: The authors have ameliorated the paper, however, I still have some points not fully addressed:

1  In the new methods section the authors should state how many papers they included in the analysis, and proper add them in the references list

Thank you for the insightful comments. We add the number of papers in the methods section (page 2). All of these articles were listed in the references.

2 Limitations and strengths should appear at the end of the discussion

Thank you very much for this important suggestion.  We have added the part about limitations and strengths at the end of the discussion on page 7.

This review focuses on the gait and speech dysfunction in PD as these symptoms cannot be well controlled by medications in a fair amount of PD patients. We included important related literature published in PubMed but not in other databases. Also, since some studies might use different connotations of the term “music therapy”, we were una-ble to identify these publications to some extent. Thus, it is likely that not all clinical stud-ies were analyzed in this review.  Our group will continue to search more databases and provide another updated review for the readers in the near future.

Thank you for allowing us to make changes,

Wei Hu on behalf of the other co-authors

Reviewer 2 Report

Comments and Suggestions for Authors

 Congratulations for your article! 

The article is relevant for the literature and also is significant for the field and for the readers. 

Author Response

(The authors gave the same response as above.)

Reviewer 3 Report

Comments and Suggestions for Authors

The authors have ameliorated the paper, however, I still have some points not fully addressed:

- In the new methods section the authors should state how many papers they included in the analysis, and proper add them in the references list

- Limitations and strengths should appear at the end of the discussion

Author Response

(The authors gave the same response as above.)
